# Predicting a Fall Based on Gait Anomaly Detection: A Comparative Study of Wrist-Worn Three-Axis and Mobile Phone-Based Accelerometer Sensors

**DOI:** 10.3390/s23198294

**Published:** 2023-10-07

**Authors:** Primož Kocuvan, Aleksander Hrastič, Andrea Kareska, Matjaž Gams

**Affiliations:** 1Department of Intelligent Systems, Jožef Stefan Institute, 1000 Ljubljana, Slovenia; 2Faculty of Electrical Engineering (FE), 1000 Ljubljana, Slovenia; ah3001@student.uni-lj.si; 3Faculty of Veterinary Medicine, 1000 Ljubljana, Slovenia; ak2869@student.uni-lj.si

**Keywords:** personalized, supervised learning, three-axis accelerometer, ambient intelligence, elderly people, gait abnormalities, predicting falls, accelerometer features, PCA, t-SNE

## Abstract

Falls by the elderly pose considerable health hazards, leading not only to physical harm but a number of other related problems. A timely alert about a deteriorating gait, as an indication of an impending fall, can assist in fall prevention. In this investigation, a comprehensive comparative analysis was conducted between a commercially available mobile phone system and two wristband systems: one commercially available and another representing a novel approach. Each system was equipped with a singular three-axis accelerometer. The walk suggestive of a potential fall was induced by special glasses worn by the participants. The same standard machine-learning techniques were employed for the classification with all three systems based on a single three-axis accelerometer, yielding a best average accuracy of 86%, a specificity of 88%, and a sensitivity of 86% via the support vector machine (SVM) method using a wristband. A smartphone, on the other hand, achieved a best average accuracy of 73% also with an SVM using only a three-axis accelerometer sensor. The significance analysis of the mean accuracy, sensitivity, and specificity between the innovative wristband and the smartphone yielded a *p*-value of 0.000. Furthermore, the study applied unsupervised and semi-supervised learning methods, incorporating principal component analysis and t-distributed stochastic neighbor embedding. To sum up, both wristbands demonstrated the usability of wearable sensors in the early detection and mitigation of falls in the elderly, outperforming the smartphone.

## 1. Introduction and Related Work

Falls are a major health concern as well as a significant cause of injury among the older population, with an estimated one in four individuals aged 65 and above experiencing a fall each year [1]. The consequences of falling can extend beyond physical injury in the form of reduced mobility, a decreased ability to perform daily activities, an increased burden on caregivers, or even mortality [2,3]. A fall can precipitate a range of psychological consequences, such as post-fall anxiety syndromes, a fear of falling, a diminution in self-efficacy, a reduction in mobility, and decreased levels of social engagement, leading to a lower quality of life [4]. This highlights the urgency of developing affordable systems that can reliably predict falls and thus provide warning and assistance prior to the actual fall.

Research has indicated that falls involving older adults result from the complex interactions between intrinsic and extrinsic factors, including cognitive impairment, sensory deficits, mobility limitations, medication use, and environmental risks [5]. Among these factors, balance deficit and gait impairment are frequently identified as key contributors to the risk of a fall. Gait abnormalities can often serve as an early indicator of underlying medical conditions, making them a potentially valuable diagnostic tool for predicting future disease progression, as well as future falls [6]. Furthermore, a gait analysis can be used to evaluate the seriousness and extent of medical conditions, to track intervention outcomes, and to forecast the intervention’s effectiveness. This is particularly important when seeking relevant information about the progression of various illnesses, including neurological diseases like multiple sclerosis or Parkinson’s, systemic disorders such as cardiac conditions that affect gait, orthopedic diseases, and age-related diseases [7]. Consequently, fall prevention has become an important area of research in healthcare [8].

Several systems have been developed to address this issue by focusing on detecting falls and notifying the person’s contacts once a fall occurs. An alternative is developing fall prediction and fall prevention systems that can predict and prevent falls by using wearable sensors to gather gait data. The data are then analyzed with machine-learning (ML) algorithms to predict the risk of a fall in the future. Both fall prediction and fall prevention systems can be very important because they evaluate fall risk and enable recovery mechanisms to be implemented before the fall actually occurs [9].

There are two primary approaches to examining human gait using technology: non-wearable and wearable systems. Non-wearable systems take place in controlled research facilities where the sensors capture the gait data while the person walks on a designated walkway. Wearable sensors, on the other hand, allow for the analysis of gait data outside of laboratory settings. Inertial sensors like accelerometers are used in wearable devices to measure an object’s velocity, acceleration, orientation, and the gravitational forces. Accelerometers use Newton’s Law of Motion to calculate the acceleration by factoring in a known mass and the measured forces. The development of miniaturized sensors and wireless communication systems has made it possible to obtain real-time measurements of gait during everyday activities by attaching devices to different parts of the body [7].

Wrist-worn accelerometers have the potential to provide important information that could predict and detect falls in real time [10]. Recent technological advancements, including larger memory capacities, wider acceleration ranges, smaller sizes, and lower cost, have made accelerometers a popular alternative for use in fall detection and prevention over other body-mounted sensors. The convenience of wrist-worn accelerometers further adds to their popularity, as they could lead to higher levels of adherence in users during periods of extended use [11]. These sensors offer a non-invasive, non-intrusive, and continuous method for monitoring movement patterns, allowing the detection of subtle changes in gait that could indicate an increased risk of falling. Some recent methods have been developed to estimate gait patterns based on a single wrist-worn sensor by extracting features from the raw sensor data based on time, frequency, and statistics [12]. This includes features based on intensity, posture, periodicity, and non-gait dynamicity, as well as statistical features, such as the mean, standard deviation, and median of the acceleration norm [13]. However, wrist-worn sensors present challenges when it comes to accurately detecting gait changes due to the lack of a fixed position relative to the user’s center of mass [14]. Moreover, due to manual dexterity and frequent arm swings, the wrist habitually has movements that are independent of the gait, which represents a problem for accurate gait detection. Research has shown that wrist-accelerometer data produce somewhat worse classification results than chest- and hip-accelerometer data [15].

Advancements in multimodal (multi-sensor) wearable technologies have significantly impacted the domains of human activity recognition (HAR), fall detection (FD), and fall recognition (FR). These systems are often characterized by their simplicity, practicality, high recognition accuracy, and real-time performance capabilities. It is important to note that multimodal-based FD research is essentially a subset of HAR/FR studies, particularly when gait analysis is incorporated. However, FD focuses on a more specialized task and does not necessitate the recognition of various fall types. Recent publications illustrate the versatility of multimodal approaches in these areas. For instance, a study released this year, titled “On a Real Real-Time Wearable Human Activity Recognition System”, offers a robust solution that employs multimodal data for real-time HAR, a methodology that is directly applicable to FD as well [16]. Another recent contribution focuses on high-level feature extraction for multimodal HAR and FR, further elaborating on the state-of-the-art techniques [17]. Moreover, the constant evolution of sensing and recognition technologies necessitates ongoing attention to emerging methodologies. This is highlighted in publications such as “Sensor-Based Human Activity and Behavior Research: Where Advanced Sensing and Recognition Technologies Meet,” which underscores the importance of integrating advanced sensing capabilities into current recognition frameworks [18].

Contemporary mobile phones frequently incorporate an extensive suite of up to 20 sensors. However, it is observed that these devices are not often carried by individuals in residential settings. More importantly, our primary comparative analysis is concentrated on the functionality of a single accelerometer across diverse devices, enabling a precise evaluation of the efficacy and constraints in the context of fall prediction.

The power consumption of wearable devices is a crucial aspect to consider in the design of wearable sensor technologies. Continuous monitoring and data collecting require significant amounts of energy, which can shorten the battery life of the device. This can lead to reduced use of the wearable devices. Accelerometers are the most energy-efficient inertial sensors and they can be used to continuously record activity without the need for a battery charge. This makes them the logical choice for wrist-based sensors that can be used for long-term activity monitoring [19].

The potential for bias in the data and models used for fall detection and gait analysis is also to be examined. Bias can arise from various sources, such as differences in sensor placement, data collection methods, or the demographic characteristics of the study population. It is of great importance to neutralize the sources of bias to ensure that the models are accurate and applicable to diverse populations. Additionally, extracting single-time discrete variables from time-continuous gait data discards a lot of information and may not accurately represent the complexity of the human gait. Pre-selected variables can introduce bias and overlook important interactions between the gait characteristics and the conditions that affect the gait. To address these issues, multivariate statistical analysis and machine-learning techniques such as artificial neural networks, decision trees, and support vector machines (SVMs) have been used to objectively identify different types of gaits [20]. Also, other researchers focus on improving human activity recognition through HMM-based sequential modeling, and, e.g., in Hui Liu’s dissertation, a novel activity modeling method called Motion Units (MUs) to enhance performance is proposed. It also contributes by developing an HAR research pipeline, creating the Activity Signal Kit (ASK) software for data collection, and implementing both offline and real-time HAR systems based on multimodal biosignal datasets from over 25 subjects [21,22,23,24]. His paper on the activity modeling method for human activity recognition, utilizing Motion Units to create an operable, universal, and scalable human activity dictionary, demonstrates comparable accuracy with fewer parameters [24].

Among the methods applied to fall detection and prediction, the Hidden Markov Model (HMM) is very suitable for fall detection/recognition and reaches deep learning’s performance in many publications. This is due to its inherent sequential modeling capabilities for time series, in relation to wearable multidimensional signals [21,24,25].

Furthermore, the research indicates that normalizing spatial–temporal gait data by taking into account leg length and the subject’s age when using the SVM approach improves the accuracy from 83.3% to 96.8% [22]. In addition to normalization, the need for standardization of the methods and metrics used for fall detection and gait analysis is shown in the differences in the results from various studies. Standardization can ease the comparability of findings across different studies, as well as being useful for identifying the best practices and areas for improvement in the development of wearable sensor technologies.

To better predict the risk of falling, it is important to use optimized prediction models that take into account the type and placement of wearable sensors. Several studies explore various approaches to determine the optimal protocol for assessing gait quality. Moreover, the analysis of a dual-task gait has been proposed as a potentially valuable tool for assessing the risk of falls in older adults, as it can uncover deficits in cognitive function and mobility control that may not be evident during normal walking. This approach involves evaluating a person’s gait while they perform a secondary cognitive task, such as counting backward or reciting the alphabet, and comparing the results to those obtained during normal walking. However, there is no consensus on whether a dual-task assessment is better at predicting a fall risk than assessing a gait during single-task activities [23].

An alternative approach could be to expand the evaluation of gait quality beyond straight walking and include assessments of turning. Most studies of fall-risk prediction only focus on straight-line walking, so it might also be useful to study gait during turning movements, as turns require additional coordination and control compared to walking in a straight line. Elderly individuals at high risk of falling tend to perform turns differently from those at low risk of falling, making it possible to distinguish between what is considered a normal gait and a gait that shows signs of potential falls using turn-based data. Researchers have developed a method for classifying high-fall-risk gaits based on wearable sensor data collected during walking turns. The findings suggest that turn data contain valuable information that can improve fall-risk classification, although combining straight and turn-based features did not improve the classification models [26].

Another challenge in gait analysis is the unsoundness of the current methods under unsupervised or real-world conditions. Clinical fall-risk assessments are typically performed in controlled laboratory settings and are limited by testing locations, frequency, cost, and professional supervision [2]. Most methods have only been tested in supervised or semi-supervised settings, which do not accurately reflect the self-initiated and purposeful nature of a gait [19]. Studies have shown that there are notable differences in gait speed when comparing measurements taken in laboratory environments versus those taken during daily activities. Gait speed is said to be slower during real-world activities than during laboratory testing, according to the research [27,28]. Wearable devices offer an alternative for fall-risk assessments, as they allow the real-time monitoring of a gait in real-life conditions, at a lower cost and with minimal discomfort to the user. However, developing a method for wrist-worn devices is challenging and has not yet been validated for fall-risk assessments [2].

The objective of this paper is to compare a commercial device with a novel wristband, and both primarily with a mobile phone that has the same fall-predicting method in the task of recognizing abnormal gait patterns using a single, three-axis, wrist-worn accelerometer. Raw accelerometer data features are extracted and inputted into different types of machine-learning models. The approach used involves gathering data, extracting meaningful features, selecting the most relevant characteristics, and training a classification model to differentiate a normal from an abnormal gait. The performance of this approach is assessed using a dataset that includes parameters of the normal and abnormal walking patterns of 17 test subjects.

## 2. Dataset

### Performing the Experiment and Data Collection

In light of the unavailability of adequate datasets featuring inertial sensors for wrist-worn devices, a novel dataset was created for public use. The dataset was generated using an experiment conducted within a confined room, where the participants traversed a predefined loop. Two wristbands, namely, the Caretronic wristband and the Empatica E4 wristband, were worn by each participant during the experiment. The Caretronic wristband utilized the Hometab device, connected to the internet via a Bluetooth connection, to stream data to a Firebase server. On the other hand, the Empatica E4 wristband employed a smartphone application to transmit data to its dedicated cloud service.

The designated walking route encompassed a level surface with no inclinations, primarily straight with the exception of a concluding semi-circular section used for turning around. The obstacle of the path can be seen in Figure 1, whereas the walk around the angle was substantially more curved. This setup facilitated the acquisition of uninterrupted gait data over extended periods, akin to real-world scenarios encountered during daily walks. To emulate the walking pattern of elderly individuals, each participant wore weighted leg attachments, amounting to 3 kg per leg. In the context of our study, we utilized vision impairment goggles [29] (illustrated in Figure 2b) as a means to simulate a deteriorated gait. These specialized goggles were engineered to induce a visual perception mimicking the experiential semblance of an individual under the influence of alcohol when peering through these devices. At the onset of the experimental protocol, a significant proportion of the subjects displayed marked challenges in their ability to walk. However, following a period of persistent goggle usage, the subjects demonstrated an adaptation to their visually altered environment, seen as an improvement in their locomotion over the course of the experiment. Despite the observed acclimation to the conditions imposed by the goggles, the majority of the subjects’ locomotive patterns continued to show signs of abnormality, indicating that their gait remained not entirely normal.

The experiment for each of the participants was split into 8 sessions. Initially, the participants walked without impaired vision for a period of 8 min, following a clockwise direction for the initial 4 min and subsequently an anti-clockwise direction for the remaining 4 min. Subsequently, the position of the wristbands was altered, and the procedure was repeated. Lastly, all the aforementioned steps were replicated while the participants wore the impairment goggles. The described experiment sessions are shown in Table 1. Discrepancies observed in the signals recorded by the Caretronic wristband and the Empatica E4 wristband can be attributed to the different positions and the coordinate system orientations of the micro-electromechanical system (MEMS) chip housing the accelerometer on the wristbands (Figure 3). On the other hand, the differences in the signals with and without goggles are not visually detectable to humans, while the differences in an actual walk usually can be differentiated when observing the same person.

In the conducted study, we selected the Xiaomi Redmi 7 Smartphone as the experimental apparatus. It is postulated that any device from the same manufacturer, or equivalently sophisticated models produced within the preceding decade, would likely yield analogous outcomes. During the experimental process, the smartphone was housed within a compact pouch that the participants wore around their midsections. Consequently, the motion of the pouch did not resemble that of a bag oscillating in hand, analogous to wristband movement, but rather dangled from a shoulder, exhibiting greater movement than if fastened directly to a belt.

The Caretronic wristband (Figure 4a) is equipped with an ARM Nordic NRF52840 microcontroller. This microcontroller interfaces with both the gyroscope and accelerometer sensors through the Two-Wire Interface (TWI). Designed as an affordable solution, the Caretronic wristband is aimed at achieving global reach, including in countries with a lower GDP. In our study, we compared the performance of the Caretronic wristband with that of the Empatica E4 (Figure 4b). The latter, while more costly, also features pulse and temperature sensors. We configured the sampling rate of the Caretronic wristband at 52 Hz, though it possesses the capability for higher rates. Conversely, the Empatica E4 maintains a fixed sampling rate of 32 Hz for its accelerometer sensor. The primary objective of this analysis is to contrast the performance metrics of the Empatica E4 and Caretronic wristbands and the smartphone using only the accelerometer sensor.

## 3. Methodology

### 3.1. CR-Features Library for Python

In our study, we employed the cr-features Python library, which was developed by the Department of Intelligent Systems at the “Jožef Stefan” Institute. This library is specifically designed for the computation of diverse accelerometer features that are well suited for context recognition applications [30].

To ensure a focused analysis, we conducted a feature selection process due to the library’s capacity to calculate a substantial number of features, specifically 169 in our case. Through this process, we carefully selected a subset of 87 features that demonstrated the highest relevance and discriminative power for our research objectives. This feature selection procedure was used to enhance the interpretability and efficiency of the subsequent analyses.

### 3.2. Semi-Supervised Learning

Dimensionality reduction is a machine-learning technique that is widely used in data analysis. The idea is the elimination of irrelevant or redundant variables while simultaneously conserving valuable information from the original dataset. It focuses on transforming high-dimensional data into a lower-dimensional representation that aligns with the intrinsic dimensionality of the data, or the minimum number of parameters needed to capture the essential characteristics of the data [31].

Having high-dimensional data can often be computationally expensive to process and analyze. Dimensionality reduction can be helpful in improving computational efficiency by identifying and eliminating redundant features, as well as features that can cause the over-fitting of the models. By doing this, an acceptable generalization capability can be achieved. High-dimensional datasets often contain noise or measurement errors, which can be filtered out by using different dimensionality reduction techniques. This can improve the overall data quality and analysis [32]. These techniques are broadly used in fields, such as machine learning and statistics. They can be categorized into feature selection, which aims to identify and obtain the most relevant variables from the original dataset, and dimensionality reduction, which creates a smaller set of new variables while keeping the same base information as the original data [33]. In the following subsections, we will briefly describe the two most important and used algorithms for dimensionality reduction.

We divided the signals obtained from the Empatica wristband and Caretronic wristband separately into 10 s, non-overlapping windows. Each window was used to calculate 169 features using the cr-features library. Following the feature calculation, we employed principal component analysis (PCA) to transform the 169-dimensional space into a 17-dimensional space. We opted for using PCA (principal component analysis) over LDA (Linear Discriminant Analysis) due to the nature of our classification problem, which is binary. According to the scikit-learn library documentation, LDA is primarily intended for multiclass problems, stating that “The dimension of the output is necessarily less than the number of classes, so this is in general a rather strong dimensionality reduction, and only makes sense in a multiclass setting.” Source: Scikit-learn LDA/QDA Documentation. Seventeen components were selected because the explained variance of the PCA was 0.95, which means that the reduced dimensional space contains a 95% variance of the original dimensional space. On the PCA-transformed data, t-SNE was applied to generate a two-dimensional scatter plot. The complete process, starting from the initial step of generating signal windows to the final step of constructing a 2D abstract scatter plot, is illustrated in the flowchart in Figure 5.

#### 3.2.1. Principal Component Analysis

Principal component analysis (PCA), dating back to 1901, is a popular linear technique for dimensionality reduction [33] in pattern recognition and compression schemes. The general idea that PCA represents is finding a new, smaller set of the most relevant variables, called principal components, that capture the maximum variance of the data [34]. By projecting the original dataset onto this principal component, PCA aims to reduce the dimensionality while keeping the most important information. It does this by creating a linear subspace of lower dimensionality that captures the essential variability in the data. A part of the process is finding a set of orthogonal directions that explain the most variance of the data and project the data onto these directions [35], as well as finding a linear mapping that maximizes the variance of the data, which is achieved by computing the eigenvalues and eigenvectors of the covariance matrix of the data. The eigenvectors represent the principal directions and the eigenvalues reveal the variance of each principal component. By selecting the top principal components with the largest eigenvalues, dimensionality reduction can be attained [31].

Additionally, PCA has the ability to convert correlated features into an uncorrelated feature vector by selecting the principal components that correspond to the highest eigenvalues [36]. Because the reduced-dimensional representation can be easily visualized, PCA can also be used for data visualization.

However, PCA does have some disadvantages. Data standardization is recommended when using PCA, because it is otherwise unable to find the optimal principal components due to the sensitivity of the feature scales. Furthermore, its ability is limited to only finding a linear subspace and generally it does not perform well on nonlinear data.

#### 3.2.2. t-Distributed Stochastic Neighbor Embedding

The t-distributed stochastic neighbor embedding (t-SNE) is a nonlinear dimensionality reduction technique that works toward preserving the local and global structure of the data. While PCA operates in a linear subspace, t-SNE implements different transformations in different regions with the idea of keeping similar data points in a low-dimensional space closer together [37,38].

t-SNE uses Gaussian (normal) distributions to measure and model the similarities between high-dimensional data points as conditional probabilities. It then creates a low-dimensional map where another set of probabilities represents the similarities. The newly created map aims to match the pairwise similarities from the high-dimensional space [39].

t-SNE can capture complex nonlinear relationships, which makes it a good choice for nonlinear data visualization. Additionally, it is able to consider and represent both local and global structures in the data during the dimensionality reduction process, as well as to predict the number of each point’s close neighbors [38].

However, t-SNE is a computationally expensive algorithm, especially for large datasets, and its results can be sensitive to the choice of hyperparameters. While t-SNE is appropriate for capturing local structures and keeping pairwise similarities, it is not yet clear how it performs on general dimensionality reduction tasks. Furthermore, because t-SNE is a local-neighborhood-based method, it assumes that local similarities in the high-dimensional space can be truthfully represented in the low-dimensional space. It may not give satisfactory results in cases where the intrinsic dimensionality of the data is high or variable across the dataset. It is also important to note that t-SNE is not guaranteed to converge to a global optimum of its cost function [38].

These weaknesses highlight the importance of carefully considering the applicability and limitations of dimensionality reduction techniques in different scenarios. In terms of applications, PCA is widely used for feature extraction, data compression, noise reduction, and speeding up machine-learning algorithms. It is especially useful when the linear relationships in the data are relevant for the analysis. On the other hand, t-SNE is commonly employed for data visualization, clustering analysis, and identifying hidden patterns in the data. Its ability to capture complex structures and reveal local relationships makes it particularly valuable in exploratory data analysis.

### 3.3. Supervised Learning

Supervised learning is a sub-field of machine learning that involves labeling data and constructing a model with the guidance of a supervisor. In this study, we employed classical models, namely, K-Nearest Neighbors, support vector machine, and AdaBoost. These models were applied to the calculated 169 features, after the feature selection process. Additionally, the hyperparameters were optimized for each classifier. Within the realm of supervised learning, we assume prior knowledge of the normal and abnormal gait patterns of each individual. This approach enables us to quantitatively evaluate the performance of classical methods without preprocessing the accelerometer signals. Notably, signal preprocessing was omitted in this research to avoid impeding operations on the microcontroller, as our aim in the future is to utilize the algorithm on a low-speed microcontroller to reduce costs.

The subsequent subsections will provide concise descriptions of the machine-learning methods employed in this study.

#### 3.3.1. K-Nearest Neighbours

The K-Nearest Neighbors (K-NN) algorithm [40] is a nonparametric approach predominantly employed for classification and regression tasks. Being nonparametric, it avoids assumptions about the underlying data distribution. This algorithm utilizes the entire training dataset during prediction, making it a memory-based learning method that does not require a dedicated learning phase.

When presented with a new instance, the K-NN algorithm classifies it by identifying the “k” training instances that are the most similar to it in the feature space. The similarity is typically determined using distance measures, such as Euclidean (Equation 2) and Manhattan (Equation 1). The choice of the distance metric depends on the characteristics of the data. Selecting an appropriate number of nearest neighbors (“k”) is a critical factor in the K-NN algorithm. A small “k” value can lead to sensitivity to noise, while a large “k” value may result in an overly generalized model.
(1)$$d_{manhattan}\left(P,Q\right)=\sum_{i=1}^{n}\left|p_{i}-q_{i}\right|$$
(2)$$d_{euclidean}\left(P,Q\right)=\sqrt{\sum_{i=1}^{n}{\left(p_{i}-q_{i}\right)}^{2}}$$
where $P=(p_{1},p_{2},\ldots ,p_{n})$ and $Q=(q_{1},q_{2},\ldots ,q_{n})$ are the two points in n-dimensional space.

#### 3.3.2. Support Vector Machine

Support vector machines (SVMs) [41] are highly effective supervised learning algorithms primarily utilized for classification tasks, but they can also be applied to regression and outlier detection tasks. The core principle behind SVMs is the creation of a hyperplane that optimally separates the data points belonging to different classes. This separation aims to maximize the margin between the closest points to the hyperplane, known as support vectors, from each class.

Given a training set $\left(x_{1},y_{1}\right),\ldots ,\left(x_{n},y_{n}\right)$, where $x_{i}\in R^{p}$ and $y_{i}\in \left\{-1,1\right\}$, SVMs seek to minimize the value of ${\left||w|\right|}_{2}^{2}$. The parameters *w* and *b* are involved in the minimization process and are subject to the constraint described in Equation (Equation 3).
(3)$$y_{i}\left(w^{T}x_{i}-b\right)\leq 1$$

In scenarios where the data are not linearly separable, the SVM employs the kernel trick to map the original feature space into a higher-dimensional space where a separating hyperplane can be identified. This approach enables SVMs to effectively handle nonlinearly separable data. Rather than explicitly performing the transformation, a kernel function $K(x_{i},x_{j})$ is used to compute the dot product in the higher-dimensional space. By utilizing the kernel function, the computational efficiency of the SVM algorithm is improved. Commonly employed kernel functions include the linear, polynomial, and radial basis function (RBF) options.

#### 3.3.3. AdaBoost

AdaBoost, short for Adaptive Boosting, is an ensemble machine-learning algorithm primarily designed to enhance the performance of classification tasks by amalgamating the outputs of multiple weak learners [42].

Initially, each instance in the training set is assigned an equivalent weight. In each iterative step, a weak learner is trained. This learner prioritizes minimizing errors, particularly emphasizing instances that were misclassified in previous iterations.

Subsequent to each iteration, the algorithm computes the weight for the current weak learner’s output, based on its performance. Concurrently, it modifies the weights of training instances, incrementally augmenting the weights of the previously misclassified examples.

The salient feature of AdaBoost is that the final decision is not solely the output of an individual weak learner. It is a weighted amalgamation of the predictions from all the weak learners, where higher weights are attributed to those with superior performance. This method has gained significant traction in scenarios where the combination of multiple weak learners results in a more accurate prediction.

However, AdaBoost can encounter difficulties with noisy data or outliers that deviate significantly from the general trend. To ameliorate this, the hyperparameters of the AdaBoost algorithm should be judiciously optimized.

### 3.4. Validation and Evaluation Metrics

For evaluating the overall performance measure of the methods, various metrics were applied in this study. Considering the fairly evenly distributed dataset, the primary evaluation metric used in this study was accuracy, which is the ratio of the number of correct predictions over the total number of predictions. Additionally, the sensitivity (true positive rate) and the specificity (true negative rate) metrics were analyzed to obtain a more detailed understanding of the performance of the models.

The sensitivity was calculated as the ratio of the true positives to the sum of the true positives and false negatives, while the specificity was calculated as the ratio of the true negatives to the sum of the true negatives and false positives. The accuracy was calculated as the ratio of the true positives plus true negatives to the sum of the true positives, true negatives, false positives, and false negatives. True positives represented the number of correct positive predictions, true negatives represented the number of correct negative predictions, false positives represented the number of incorrect positive predictions, and false negatives represented the number of incorrect negative predictions.

Other metrics such as the F1-score and *FNR* were also used to evaluate the performance of the models. The F1-score is an evaluation metric that combines the precision and recall scores. The false negative rate (*FNR*) metric is calculated as 1—true positive rate (*TPR*), also known as the sensitivity. A lower *FNR* signifies improved outcomes. Given the context, a lower *FNR* is optimal for our use case as it implies fewer missed identifications of the individuals with an abnormal gait, as compared to falsely identifying those with a normal gait as abnormal. We show only the sensitivity in the tables because we can calculate the *FNR* using Equation (Equation 7).
(4)$$\mathrm{Accuracy}=\frac{TP+TN}{TP+FP+TN+FN}$$
(5)$$\mathrm{Sensitivity}=\frac{TP}{TP+FN}$$
(6)$$\mathrm{Specificity}=\frac{TN}{TN+FP}$$
(7)$$FNR=\frac{FN}{FN+TP}=1-TPR$$
(8)$$F1\;\mathrm{score}=2\times \frac{\mathrm{Precision}\times \mathrm{Recall}}{\mathrm{Precision}+\mathrm{Recall}}$$

Six configurations of the cross-validation methods were used in our analysis; each configuration separated the training and testing data into two subsets.

For the first cross-validation configuration, the training subset was the Caretronic device being on the right hand and the Empatica on the left hand with the person walking in both clockwise and counterclockwise directions and the Caretronic device on the left hand and Empatica device on the right hand with a person walking just in the clockwise direction, both with and without the impairment glasses. The second, third, and fourth configurations were also carried out in the same sense but with other combinations of sessions. In the fifth configuration, the train set consisted of all the sessions in which a person was walking in a clockwise direction and the test set in which the person was walking in a counterclockwise direction. In the sixth configuration, the train and test subsets were interchanged. All the configurations are listed in Table 2 for which the sessions are referenced in Table 1.

The cr-features were calculated and the Adaboost, SVM, and K-NN methods were used. Overall, by utilizing different metrics such as the accuracy, sensitivity, specificity, and F1-score, an in-depth evaluation of the models’ performance was completed.

## 4. Results

In this section, the results are presented based on two principal methodologies: the supervised machine-learning approach described in Section 4.1 and the scatter plot representations in Section 4.2. Each subsection details the variation observed in the gaits and their classifications.

Table A1 in Appendix A delineates the demographics and personal data of the participants. The inclusion of personal data alongside raw accelerometer data did not lead to significant improvements in the evaluation metrics of the methods employed.

### 4.1. Supervised Learning Results

The detailed findings obtained from the personalized supervised learning techniques are tabulated in Table A2 in Appendix A (column names abbreviated) and are summarized in Figure 6 for configurations 1–4 and Table 3. The results show that the Caretronic wristband using only the accelerometer sensor performs better than the Empatica E4 wristband with the SVM model, and both wristbands notably surpassed the smartphone in their respective performance metrics. This is also confirmed in Table 4 and Table 5 by a two-tailed, paired *t*-test to evaluate the accuracy, specificity, sensitivity, and F1 of the two different devices. For the SVM model, the critical t-values consistently undershot their counterparts from the Student’s distribution, a pattern also mirrored by the computed *p*-values. However, this differential in performance efficacy was not consistent for the AdaBoost and KNN models, being predominantly evident only in the SVM model. A granular breakdown, contrasting the Caretronic wristband with the smartphone, evinced values approximating 0.000.

The reasons why the Caretronic wristband outperformed the Empatica device probably relate to the design purpose. While the former is engineered for task specificity, the latter assumes a more generic, multipurpose role. A synergistic deployment of both wristbands for classification further amplifies the accuracy.

In reference to the smartphone, it should be noted that the experimental setup was conducted where each participant was given a pouch with a smartphone placed within it. In some cases, the smartphone demonstrated additional movement within the pouch, as might be common in real life. The wristbands, on the other hand, were tied to the wrist in a way no unrestricted movement was permitted. Additionally, accelerometers on wristbands may provide more information than when loosely attached to a body in a mobile phone. Both issues need further to be analyzed.

Table A3 shows the time complexity of each method for each dataset.

### 4.2. Semi-Supervised Learning Results

Two scatter plots are presented in Figure 7 and Figure 8 illustrating the gait data for subjects 16 and 11 constructed from 2D t-SNE-transformed data, following the PCA transformation to 17 dimensions from 169. These graphical representations provide a visual differentiation of the gait classes, demonstrating our semi-supervised learning approach. This segregation of classes suggests the viability of a semi-supervised learning approach. The clustering could also provide a tool for classification as in ML. However, it should be noted that Figure 7 is an example of distinctive visual separation, and Figure 8 represents an average one. There is no point in presenting most similar walks with and without goggles as there is no relevant difference.

The performance of individual test subjects varied significantly. Some were disoriented when wearing the goggles, while others remained unaffected, walking as they normally would without goggles.

In real life, the new data points incorporated enable a re-calculation of the t-SNE and the median center. This could be carried out in the 2D t-SNE space, using the Euclidean distance measure as an evaluation metric. If these newer data points begin to diverge from the calculated center, it might indicate abnormal gait classes.

## 5. Conclusions

The primary focus of this research is to contrast the accelerometer positioning on wristbands with that of a mobile phone in a pouch, with the aim of detecting degradation in gait. All approaches used the same conventional machine-learning techniques and relied on a sole 3D accelerometer. The selection of the two wristbands was informed by our familiarity with their operation and to elucidate variations inherent within the wristband methodology.

A new dataset with 17 test subjects was obtained and can be used by other researchers for benchmarking (see Data Availability Statement).

In a prior investigation [2], it was demonstrated that employing multimodal sensors in devices yielded sufficient accuracy when detecting gait abnormalities. However, our study demonstrated that even single-sensor devices can deliver commendable performance by implementing an altered algorithm comprising classical computational machine-learning methods. Our previous study differs in several ways from [2]: We developed a personalized model, while they developed a generalized model that identifies gait abnormalities in a chosen population. Second, the comparison between two wristbands and a smartphone was performed. Next, the adoption of a lone accelerometer not only reduces the costs associated with buying the device but also lowers the energy consumption, thereby enabling its utilization in low-power applications. Furthermore, the compact, light form and ease of attachment of such a device is an added advantage. Along the way, it was established that Caretronic’s affordable wristband, available at a modest price of only a few tens of EUR, achieves better accuracy than the more expensive Empatica E4 wristband in our specific scenario. However, it should be noted that the novel wristband is focused on specific purposes, while Empatica E4 is a rather general-purpose wristband. In these tests, the SVM model gave similar results to those reported in [2] with a computationally more demanding convolution neural network (CNN). Also, we noticed that combining two quality wristbands produces better results.

Several published studies report high accuracy levels exceeding 99% for tasks comparable to ours. Specifically, Thakur et al. documented a performance accuracy of 99.87% while Lee et al. reported an accuracy of 99.38% [43,44]. Nonetheless, it is crucial to acknowledge that the experimental conditions in these studies diverged from ours. For instance, even upon detailed visual analysis, it was evident that certain participants exhibited minimal gait alterations when wearing the goggles, especially as the experiment progressed. In predicting future falls, it is a well-acknowledged premise that achieving absolute perfection is unattainable; consequently, experimental designs should aim to facilitate only probabilistic predictions. Moreover, the experiment should enable differentiation and ranking of the methods.

The limitations of this study are centered around the issue of the mobile phone not being firmly attached to an individual, which the wristbands were. This was the chosen scenario, while it is assumed that the 3D accelerometer in a mobile phone when firmly attached to the wrist would achieve similar results than those of the wristbands. Therefore, a mobile phone in a pouch or a bag is one of the real-life scenarios and therefore the study represents a valid comparison under those circumstances. Also, wristbands can be loosely placed on a wrist, causing additional movement, which was not allowed in this experiment.

The second limitation is that the clustering algorithms were used only for the visual analysis, whereas they could be used for classification as well. This is also one of our future projects.

Future work should encapsulate several placements of mobile phones, applying classification procedures for clustering and combining several sensors. One of the interesting ideas is to include two accelerometers in the same wristband, because the measurements in our scenarios indicate significant improvements while ensuring low costs and energy consumption.

## Figures and Tables

**Figure 1 sensors-23-08294-f001:**
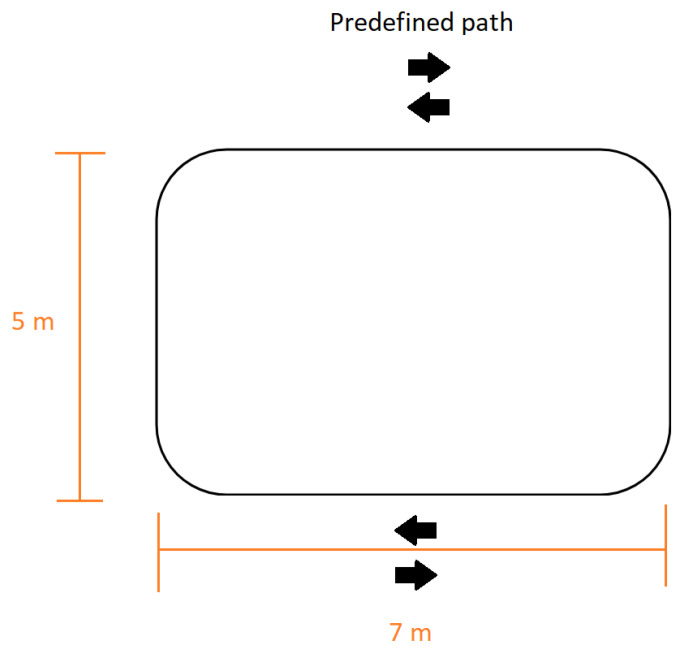
Performed predefined path simulating walk.

**Figure 2 sensors-23-08294-f002:**
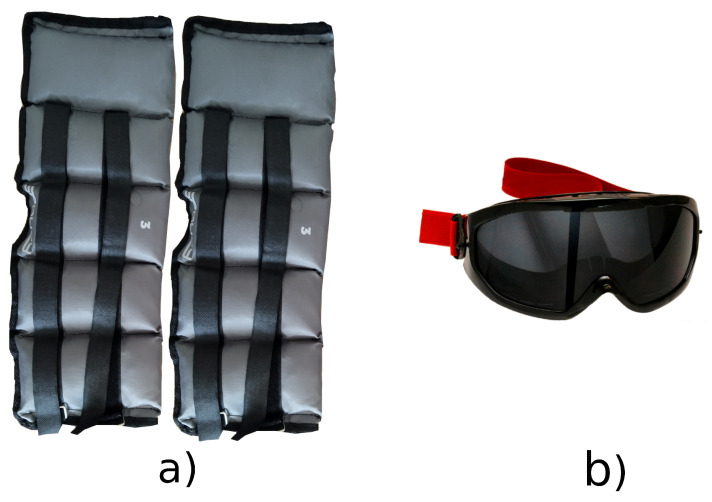
Equipment for simulating elderly people with an abnormal walk. (**a**) Leg weights—3 kg each. (**b**) Impairment glasses.

**Figure 3 sensors-23-08294-f003:**
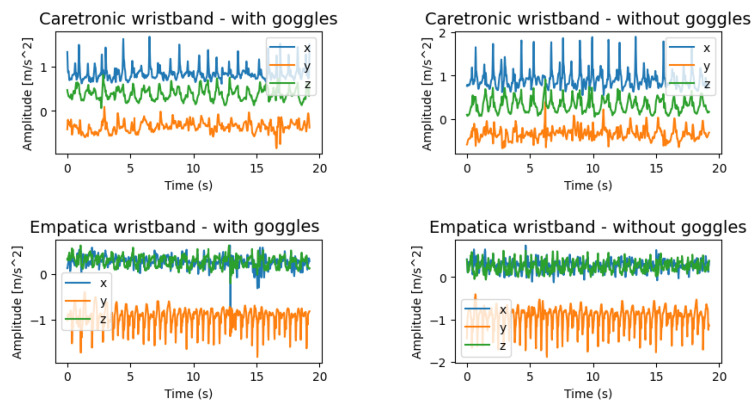
Caretronic and Empatica E4 raw accelerometer signals for x, y, and z axes.

**Figure 4 sensors-23-08294-f004:**
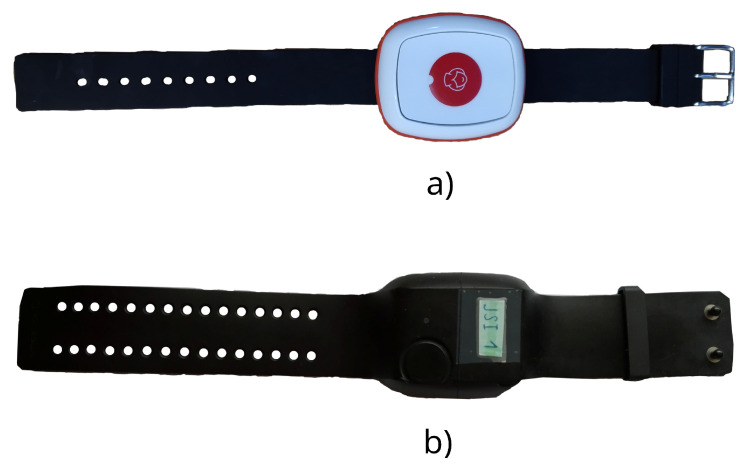
Equipment for data collection: (**a**) Caretronic wristband, (**b**) Empatica wristband.

**Figure 5 sensors-23-08294-f005:**
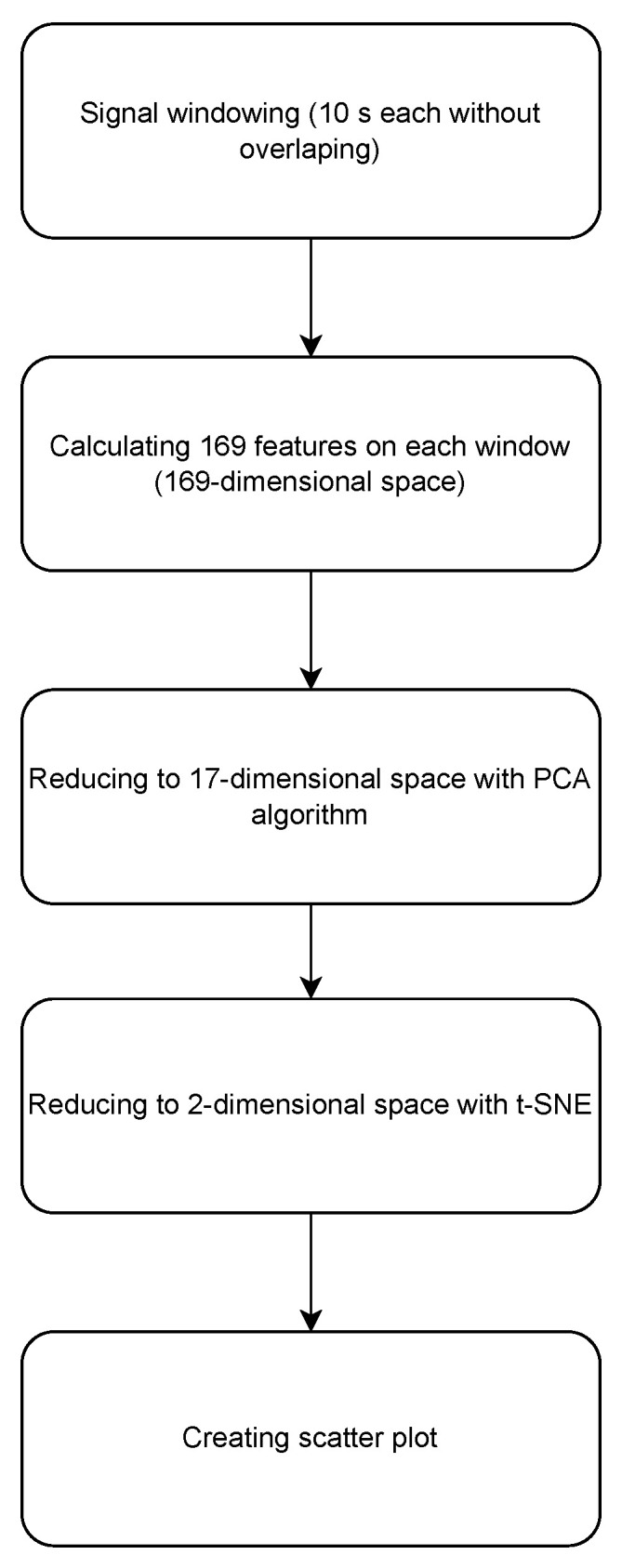
Flowchart of the proposed algorithm.

**Figure 6 sensors-23-08294-f006:**
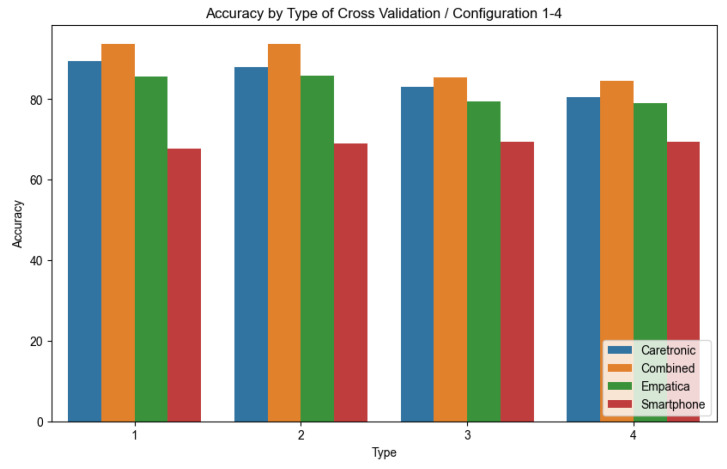
Plot shows a comparison of three devices used and a combination of features extracted from two of them (combined). The *x*-axis 1–4 corresponds to the type of cross validation from Table 2, based on the accuracy averaged over every subject in the study for the model (SVM), sorted by the descending order of the *y*-axis.

**Figure 7 sensors-23-08294-f007:**
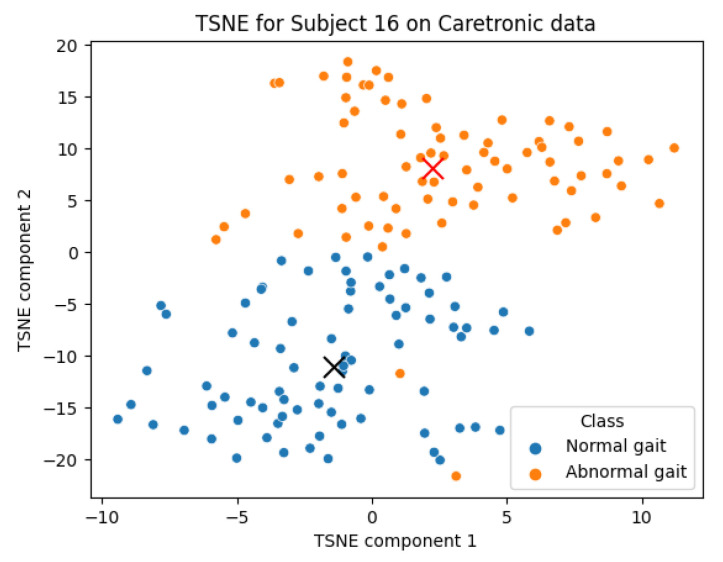
This scatter plot is a two-dimensional TSNE representation of data for subject 16.

**Figure 8 sensors-23-08294-f008:**
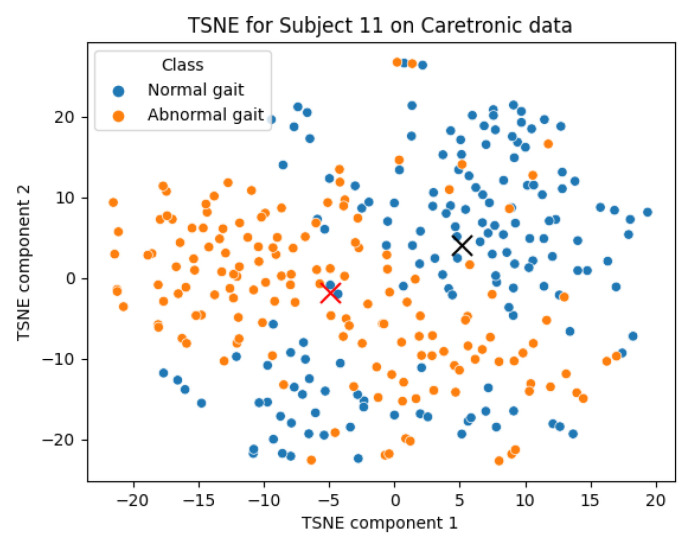
This scatter plot is a two-dimensional TSNE representation of data for subject 11.

**Table 1 sensors-23-08294-t001:** Sessions of experiment per participant.

Session	Impairment Goggles	Left Hand	Right Hand	Clockwise Walking Direction
1	No	Caretronic	Empatica	Yes
2	No	Caretronic	Empatica	No
3	No	Empatica	Caretronic	Yes
4	No	Empatica	Caretronic	No
5	Yes	Caretronic	Empatica	Yes
6	Yes	Caretronic	Empatica	No
7	Yes	Empatica	Caretronic	Yes
8	Yes	Empatica	Caretronic	No

**Table 2 sensors-23-08294-t002:** Cross-validation configurations.

Configuration	Train Sessions (in Equation (1))	Test Sessions
1	1, 2, 3, 5, 6, 7	4, 8
2	1, 2, 4, 5, 6, 8	3, 7
3	1, 3, 4, 5, 7, 8	2, 6
4	2, 3, 4, 6, 7, 8	1, 2

**Table 3 sensors-23-08294-t003:** Average results of supervised learning methods for Empatica E4, Caretronic, and smartphone. The best results are marked with **bold**.

Classifier	Device	Accuracy (%)	Sensitivity (%)	Specificity (%)	F1
ADA	Empatica	75.4	77.9	76.7	76.2
	Caretronic	**82.9**	**84**	**81.7**	**83**
	Smartphone	65	78.2	77	70.1
KNN	Empatica	77.4	83.8	73.7	78.2
	Caretronic	**82.5**	**83.4**	**83.3**	**83.3**
	Smartphone	71.9	76.7	73.5	72
SVM	Empatica	79.2	80.7	77.7	79.5
	Caretronic	**86.7**	**86.7**	**88.8**	**86.1**
	Smartphone	73.3	72.1	75.9	73.6

**Table 4 sensors-23-08294-t004:** *p*-Values for each method, comparing between Caretronic and Empatica E4 wristbands. The best results are marked with **bold**.

Method	*p*-Value for Accuracy	*p*-Value for Sensitivity	*p*-Value for Specificity	*p*-Value for F1-Score
ADA	0.06	0.15	0.18	0.06
KNN	0.08	0.92	0.03	0.07
SVM	**0.01**	**0.04**	**0.03**	**0.02**

**Table 5 sensors-23-08294-t005:** *p*-Values for each method comparing between Caretronic and smartphone Xiaomi Redmi 7. The best results are marked with **bold**.

Method	*p*-Value for Accuracy	*p*-Value for Sensitivity	*p*-Value for Specificity	*p*-Value for F1-Score
ADA	0.000	0.3	0.3	0.000
KNN	0.01	0.01	0.16	0.01
SVM	**0.000**	**0.000**	**0.01**	**0.000**

## Data Availability

The data of the test subjects (accelerations in *x*, *y*, and *z* axes) can be found at https://portal.ijs.si/nextcloud/s/NdJKXpT8WrqTqZj (accessed on 25 July 2023).

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
