# Peer review of "Predicting a Fall Based on Gait Anomaly Detection: A Comparative Study of Wrist-Worn Three-Axis and Mobile Phone-Based Accelerometer Sensors"

_sensors, 2023, doi:10.3390/s23198294_

Round 1

Reviewer 1 Report

Please see the attached pdf file.

Author Response

The manuscript under review presents a comparative study utilizing two smart wristbands equipped with accelerometer sensors to monitor human gait and predict falls using the Support Vector Machine (SVM) method. The authors achieved accuracy rates of 88% and 86% for the wristbands, respectively, while comparing their results with data obtained from a commercial smartphone.

Regarding the scope of the “Sensors” journal, it is my view that this manuscript is suitable for this journal, given its focus on sensor application in the analysis of human gait.

In terms of innovation, the manuscript may fall slightly short as it primarily employs a straightforward SVM classification analysis of accelerometer data from the two wristbands and compares it with data from smartphones. It lacks substantial novelty in terms of methodological or qualitative advancements. It would be more meaningful to explore innovation from various angles, such as improving sensor technology (for more accurate and comprehensive data), enhancing accelerometer data processing (to extract higher-quality data and eliminate noise), and optimizing algorithms (for faster processing and improved accuracy).

R: We thank for several important ideas how to improve the paper. In the short time for another submission we implemented the remarks in the paper whenever possible.

However, the manuscript does offer some insights that can be referenced or considered for future work. It highlights the advantages of using wristband accelerometers, particularly in terms of consistent sensor placement on the body compared to smartphones, which can vary in placement. It would be beneficial if the authors could provide quantitative evidence and a more in-depth explanation of why wristbands outperform smartphones in their predictions. Additionally, if the wristbands are also subject to random placement (though they are typically worn consistently), it would be essential to discuss how this variability in sensor placement affects prediction accuracy.

R: We agree that the reason for better results with wristbands is most likely because in this case the accelerometer is directly attached to a wrist while smartphone in our experiments was in a bag that itself was not firmly attached to a body, and the smartphone in the bag was not firmly attached. Those internal movements in a bag were not measured. However, we added some explanation as to why the results with the smartphone are worse.

In Table 3, while the authors have compared SVM, ADA, and KNN based on accuracy, it would be helpful to include a comparison of computational efficiency, as this is crucial in fall prediction. Are there significant differences in computation time among these methods?

R: We included information about computational efficiency in Appendix.

Furthermore, there are some minor issues that require attention:

  1. Table 1 and Table A1 have inconsistent formatting and line shading compared to other tables. It is recommended to maintain uniformity in table layout.

R: Corrected.

  1. In Figure 2, the alignment of characters (a) and (b) is not on the same horizontal line, which may create a visual imbalance. It is advisable to rearrange the images and lettering for better alignment. Maybe I'm a bit nitpicky.

R: Corrected.

  1. In Figure 4, the titles of the two lower subfigures both state "without goggles," while one should represent "with goggles". Please confirm. Also, the legend labels in the subfigures partially obscure the data curves, which should be addressed.

R: Corrected.

  1. In Table 2, there are two lines at the bottom of the table. Typically, tables are presented with three horizontal lines. Please confirm if the same issue exists in other tables.

R: Corrected.

  1. In Table 3, while ACC, SENS, SPEC, and F1 are commonly understood abbreviations, it is advisable to annotate them with their full names for the sake of clarity and precision in the paper.

R: Corrected.

Reviewer 2 Report

Subject: Review of Manuscript Submission sensors-2616160

I have carefully reviewed the manuscript titled "Predicting a Fall Based on Gait-Anomaly Detection: A Comparative Study of Wrist-Worn 3-Axis and Mobile-Phone-Based Accelerometer Sensors" submitted to Sensors for consideration. This manuscript develops a fall-prevented alert system composed of a single, 3-axis accelerometer worn on the wrist versus a mobile phone. And authors applied unsupervised and semi-supervised learning methods, incorporating principal component analysis and t-distributed stochastic neighbor embedding, to analyze the falling risk. It cannot be accepted unless the major concerns listed below are addressed.

1.       The authors chose three commercial devices, “Empatica”, and” Caretronic”. May I know the reason for the author's choice? Why not choose an Apple or Samsung watch? Maybe they have better prediction accuracy.

2.       It noted that the author adopted two wristwatches and a smartphone to collect data. However, do old people carry two watches? This seems inconsistent with practical application.

3.       Authors claimed that they calculated 169 features on each window, and then reduced to 17-dimensional space with PCA algorithm. However, the PCA algorithm usually needs to be reduced to 5 dimensions, preferably 2 dimensions, to be a better algorithm prediction. The results of 17 dimensions are so crude that we have reason to suspect that the results are implausible. The author should explain it.

4.       Figure.6 should be exhibited by bar charts or bar diagrams, not line charts. Since the X-axis “type of cross-validation” is independent of each other. Line charts would mislead the readers.

5.       Figure.7 and 8 show the significant difference among the subjects, Figure.7 is an example of distinctive visual separation and Figure.8 represents an average one. It is commendable to show the unsatisfactory experimental data in the experiment. However, the authors should analyze the reasons for this phenomenon, or at least a predictive analysis.

6.       Table A2 lists all results gathered on one massive table. Whereas the combined results featured with ACC, SENS, SPEC, and F1 are missed. Authors should supplement these data.

Moderate editing of the English language is required.

Author Response

I have carefully reviewed the manuscript titled "Predicting a Fall Based on Gait-Anomaly Detection: A Comparative Study of Wrist-Worn 3-Axis and Mobile-Phone-Based Accelerometer Sensors" submitted to Sensors for consideration. This manuscript develops a fall-prevented alert system composed of a single, 3-axis accelerometer worn on the wrist versus a mobile phone. And authors applied unsupervised and semi-supervised learning methods, incorporating principal component analysis and t-distributed stochastic neighbor embedding, to analyze the falling risk. It cannot be accepted unless the major concerns listed below are addressed.

R: We thank for several important ideas how to improve the paper. In the short time for another submission we included as many recommendations in the paper as time allowed.

1. The authors chose three commercial devices, “Empatica”, and” Caretronic”. May I know the reason for the author's choice? Why not choose an Apple or Samsung watch? Maybe they have better prediction accuracy.

R: The reasons are as follows:
- The smartphone was put in a bag in order to compare performance not directly with a mobile phone in hand or firmly attached to a body, but to enable comparison with a smartphone loosely put in a bag, as might be a common case.

- The Caretronic wristband is a research product by our cooperating SME specialized for detecting issues with walking.

- Empatica was chosen because it is considered one of the best test wristbands for these tasks and is commonly chosen for similar tasks also at our Institute.

- It would probably be better to include Apple or Samsung watch as well in the tests, however one of the reasons for not doing so was our intention to compare methods and approaches in a scientific way and not as evaluation of commercial products.

This argumentation is included in the paper.

2. It noted that the author adopted two wristwatches and a smartphone to collect data. However, do old people carry two watches? This seems inconsistent with practical application.

R: The reasons for conducting this experiment was research curiosity. According to the principle of multiple knowledge or knowledge fusion, the two wristwatches should enable better information than the best one, and indeed this was the case. The practical meaning is somehow unclear, agreed.

3. Authors claimed that they calculated 169 features on each window, and then reduced to 17-dimensional space with PCA algorithm. However, the PCA algorithm usually needs to be reduced to 5 dimensions, preferably 2 dimensions, to be a better algorithm prediction. The results of 17 dimensions are so crude that we have reason to suspect that the results are implausible. The author should explain it.

This is explained in the paper. We follow the guidelines.

4. Figure.6 should be exhibited by bar charts or bar diagrams, not line charts. Since the X-axis “type of cross-validation” is independent of each other. Line charts would mislead the readers.

R: Accepted and included.

5. Figure.7 and 8 show the significant difference among the subjects, Figure.7 is an example of distinctive visual separation and Figure.8 represents an average one. It is commendable to show the unsatisfactory experimental data in the experiment. However, the authors should analyze the reasons for this phenomenon, or at least a predictive analysis.

R: Individually, the test subjects performed very different. Some were confused when using goggles and others were totally unphased, walking as previously without goggles. There is no point showing two nearly identical walkings in case the goggles had no effect on a test subject, therefore we added an explanation of this kind instead of another figure. This is added in the paper now.

6. Table A2 lists all results gathered on one massive table. Whereas the combined results featured with ACC, SENS, SPEC, and F1 are missed. Authors should supplement these data.

R: Table 2 was modified accordingly.

Comments on the Quality of English Language: Moderate editing of the English language is required.

R: We checked our English with ChatGPT and then hired a professional lecturer of high quality and price for the first submission. Not clear.

Reviewer 3 Report

The work contributes to the research field of fall detection. My concerns:

Why use PCA instead of LDA which is widely adopted on HAR/FD tasks? Your datas has labels.

Eq 4-8: Please strictly distinguish variables and names (\text) in formulas.

L82-84: It's not all about that. [12] is four-year-old literature. A thorough consideration of the latest technological developments is necessary. Multi-sensor wearable technologies have made great strides in the fields of human activity recognition (HAR), fall detection (FD), fall recognition (FR), etc., and their systems and methods tend to be simple and practical with good recognition rates and real-time performance. Note that multiple-sensor-based FD is essentially research that falls under the HAR/FR category (even more so when incorporating gait analysis), but FD has a simpler task and does not need to recognize various types of fall. For example, this year's On a Real Real-Time Wearable Human Activity Recognition System provides a practical, comprehensive solution of multiple-sensor activity sensing and real-time HAR, which is totally applicable to FD. On multiple-sensor HAR and FR, two month ago there is a novel work on high-level features: https://doi.org/10.1007/978-3-031-38854-5_8. In addition, new sensing+recognition technologies are also always needed to be paid attention: Sensor-Based Human Activity and Behavior Research: Where Advanced Sensing and Recognition Technologies Meet.

"Multiple-sensor" is OK but may not be a decent expression. Try "multimodal".

L84: on which basis did you say "up to 10 sensors"? Literature, or you count in your own mobile phones? Let me count for you: accelerometer, gyroscope, Magnetometer, barometer, microphone (i.e., acoustic sensor), video camera (optical sensor), light optode, thermometer, capacitive (tactile) sensor, distance sensor, fingerprint sensor, *compass, *eye tracking, *BVP, GNSS sensor (GPS), NFC sensor, *HRV, *Hall sensor, *SpO2 (blood oxygen) sensor... Even the most simple mobile phone from 2020 can contain nearly 20 sensors. And I check my list, and find that I can not confidently say which of the above mentioned sensors are definitely not beneficial for fall detecton.

L101-103. Again, you omitted some key technologies here. Also [17] is four years old. E.g., Hidden Markov Model (HMM) is very suitable for fall detection/recognition, and even reaches or exceeds deep learning's performance in many literatures, but HMM has simplicity, generalizability, and interpretability. This is due to its inherent sequential modeling capabilities for time series (for FD: wearable multidimensional signals). SOTA summarization in 2022: https://doi.org/10.1007/978-981-19-0390-8_108. Also please check "Biosignal Processing and Activity Modeling for Multimodal Human Activity Recognition" and "Motion Units: Generalized sequence modeling of human activities" for their efficient and effective HMM-based modeling for fall detection/fall recognition part.

L136-140. The up-to-date (2022) work on accurate gait speed analysis, for forward walking, walking with left/right turning, walking upstairs/downstairs, jogging, lateral walking, turning around, v-cut steping... is How Long Are Various Types of Daily Activities? It has two essential findings: (1) the duration of each healthy person's single daily motion (gait) obeys a normal distribution; (2) one single motion (such as jumping and sitting down) or one cycle in the activities of cyclical motions(such as one gait cycle in various types of walking) has an average duration in the interval from about 1-2 secs. They should be very referential for your work and ML modeling.

Author Response

Why use PCA instead of LDA which is widely adopted on HAR/FD tasks? Your datas has labels.

R: We thank for several important ideas how to improve the paper. In the short time for another submission we included as many recommendations in the paper as time allowed.

The explanation is provided in the paper now, consistent with the guidelines.

Eq 4-8: Please strictly distinguish variables and names (\text) in formulas.

R: Modified accordingly.

L82-84: It's not all about that. [12] is four-year-old literature. A thorough consideration of the latest technological developments is necessary. Multi-sensor wearable technologies have made great strides in the fields of human activity recognition (HAR), fall detection (FD), fall recognition (FR), etc., and their systems and methods tend to be simple and practical with good recognition rates and real-time performance. Note that multiple-sensor-based FD is essentially research that falls under the HAR/FR category (even more so when incorporating gait analysis), but FD has a simpler task and does not need to recognize various types of fall. For example, this year's On a Real Real-Time Wearable Human Activity Recognition System provides a practical, comprehensive solution of multiple-sensor activity sensing and real-time HAR, which is totally applicable to FD. On multiple-sensor HAR and FR, two month ago there is a novel work on high-level features: https://doi.org/10.1007/978-3-031-38854-5_8. In addition, new sensing+recognition technologies are also always needed to be paid attention: Sensor-Based Human Activity and Behavior Research: Where Advanced Sensing and Recognition Technologies Meet.

R: We studied the mentioned literature, added it in the reference lists and in the related work.

"Multiple-sensor" is OK but may not be a decent expression. Try "multimodal".

R: Included.

L84: on which basis did you say "up to 10 sensors"? Literature, or you count in your own mobile phones? Let me count for you: accelerometer, gyroscope, Magnetometer, barometer, microphone (i.e., acoustic sensor), video camera (optical sensor), light optode, thermometer, capacitive (tactile) sensor, distance sensor, fingerprint sensor, *compass, *eye tracking, *BVP, GNSS sensor (GPS), NFC sensor, *HRV, *Hall sensor, *SpO2 (blood oxygen) sensor... Even the most simple mobile phone from 2020 can contain nearly 20 sensors. And I check my list, and find that I can not confidently say which of the above mentioned sensors are definitely not beneficial for fall detecton.

R: Thank you for the remark. Even though we never used more that 10 sensors for the walking tests we agree that this number should be modified to 20. Corrected.

L101-103. Again, you omitted some key technologies here. Also [17] is four years old. E.g., Hidden Markov Model (HMM) is very suitable for fall detection/recognition, and even reaches or exceeds deep learning's performance in many literatures, but HMM has simplicity, generalizability, and interpretability. This is due to its inherent sequential modeling capabilities for time series (for FD: wearable multidimensional signals). SOTA summarization in 2022: https://doi.org/10.1007/978-981-19-0390-8_108. Also please check "Biosignal Processing and Activity Modeling for Multimodal Human Activity Recognition" and "Motion Units: Generalized sequence modeling of human activities" for their efficient and effective HMM-based modeling for fall detection/fall recognition part.

R: We have studied the mentioned literature, added in the literature list and shortly described it in the related work.

L136-140. The up-to-date (2022) work on accurate gait speed analysis, for forward walking, walking with left/right turning, walking upstairs/downstairs, jogging, lateral walking, turning around, v-cut steping... is How Long Are Various Types of Daily Activities? It has two essential findings: (1) the duration of each healthy person's single daily motion (gait) obeys a normal distribution; (2) one single motion (such as jumping and sitting down) or one cycle in the activities of cyclical motions(such as one gait cycle in various types of walking) has an average duration in the interval from about 1-2 secs. They should be very referential for your work and ML modeling.

R: Agreed that the specific movements like sitting or standing up are important indicators for potential warning, however, we concentrated on walking as it is commonly more informative as other types of movements. In addition, falls happen mainly to elderly and in most cases, elderly are not healthy persons. At the moment we are not sure how to improve this valuable comment in the research, but thank for all the valuable remarks.

Reviewer 4 Report

The following are my comments for improvement of this paper:

1. Several fact-based statements throughout the paper are missing supporting references. For instance, this statement – “A fall can precipitate a range of psychological consequences, such as post-fall anxiety syndromes, a fear of falling, a diminution in self-efficacy, a reduction in mobility and decreased levels of social engagement, leading to a lower quality of life” should have a supporting reference.

2. The authors state that the accuracy of their model is 86%. How is this an improvement over two similar works that already exist? Specifically, Thakur et al. (see - https://doi.org/10.3390/jsan10030039) achieved a performance accuracy of 99.87%, and Lee at el. (see - https://doi.org/10.1109/JSEN.2019.2918690) achieved performance accuracy of 99.38%. 

3. The authors state – “The dataset was generated using an experiment conducted within a confined room, where participants traversed a predefined loop” Please elaborate on the participant recruitment process. What were the inclusion and exclusion criteria for the recruitment of participants?

4. The step-by-step experimental protocol that each participant followed should be clearly presented

5. What were the diversity characteristics (for instance – age and gender) of these participants? Did diversity have any effect on the participation or performance of these participants during the experiments? Did diversity have any effect on the performance of the proposed approach for the participants?

Author Response

1. Several fact-based statements throughout the paper are missing supporting references. For instance, this statement – “A fall can precipitate a range of psychological consequences, such as post-fall anxiety syndromes, a fear of falling, a diminution in self-efficacy, a reduction in mobility and decreased levels of social engagement, leading to a lower quality of life” should have a supporting reference.

R: We thank for several important ideas how to improve the paper. In the short time for another submission we included as many recommendations in the paper as time allowed.

We included more literature to support several fact-based statements.

2. The authors state that the accuracy of their model is 86%. How is this an improvement over two similar works that already exist? Specifically, Thakur et al. (see - https://doi.org/10.3390/jsan10030039) achieved a performance accuracy of 99.87%, and Lee at el. (see - https://doi.org/10.1109/JSEN.2019.2918690) achieved performance accuracy of 99.38%.

R: This issue bothered us for quite some time so we intensely observed the data and live performance on the video. For example, some of the test subjects walked with goggles practically the same as without goggles. How is it possible to achieve nearly 100% in this case? Obviously, the tests were performed in different conditions and under different assumptions. Which of them is more relevant for real life needs to be further elaborated, or rather – our tests are probably relevant indicators for the chosen circumstances and others tests for theirs.

We discuss this issue in the paper in the discussion section.

3. The authors state – “The dataset was generated using an experiment conducted within a confined room, where participants traversed a predefined loop” Please elaborate on the participant recruitment process. What were the inclusion and exclusion criteria for the recruitment of participants?

R: We recruited test subjects on a volunteering base at our department mainly. The exclusion criteria for participants was any physical or mental issue mentioned in the prior discussion including age and overall health. Another exclusion criterion would be any problem when actually wearing googles, however, all participants had no problems with it. This was one of the reasons for choosing healthy young individuals and attach weights to approximate behaviour of elderly. In addition, the volunteers had to sign the agreement.

4. The step-by-step experimental protocol that each participant followed should be clearly presented

R: We added that in the paper.

5. What were the diversity characteristics (for instance – age and gender) of these participants? Did diversity have any effect on the participation or performance of these participants during the experiments? Did diversity have any effect on the performance of the proposed approach for the participants?

R: This is also added in the supplementary material. No major difference detected.

Round 2

Reviewer 1 Report

The manuscript has been improved.

Reviewer 2 Report

I am satisfied with the revision. It can be published in its present form. 

Reviewer 3 Report

The authors addressed my concerns well.

Still, the explanation of not using LDA is weak (a common belief rather than a scientific claim). But I think this harms less on the method itself.

I argue to accept the manuscript.

Reviewer 4 Report

The authors have revised their paper as per all my comments and feedback. I do not have any additional feedback at this point. I recommend the publication of the paper in its current form.